# Research on the Configuration Optimization of All-Metal Micro Resonant Hemisphere

**DOI:** 10.3390/s24227132

**Published:** 2024-11-06

**Authors:** Xibing Gu, Zhong Su, Xiangxian Yao, Sirui Chu

**Affiliations:** 1Beijing Key Laboratory of High Dynamic Navigation Technology, Beijing Information Science and Technology University, Beijing 100192, China; 2023020390@bistu.edu.cn (X.G.); 2023020388@bistu.edu.cn (X.Y.); 2023020380@bistu.edu.cn (S.C.); 2Key Laboratory of Modern Measurement and Control Technology, Ministry of Education, Beijing Information Science and Technology University, Beijing 100192, China

**Keywords:** metal resonator structure, positional error, frequency splitting, finite element simulation

## Abstract

As the core component of the all-metal micro resonant gyroscope, the structural parameters and form and position errors of the resonator significantly influence its vibration characteristics, and consequently, the accuracy of the gyroscope. By establishing the finite element model of an ideal hemispherical resonator and optimizing the meshing method, we refined the frequency difference to 0.1 Hz, enhancing the accuracy of the simulation model. Through finite element simulation, we examined the impact of various structural parameters and processing errors on the natural frequencies of each mode. We analyzed how form and position errors, including shell thickness error, central axis error, equatorial plane error, and edge rectangular tooth position error, affect the frequency splitting of the resonator. We provided optimization suggestions for the structural parameters, ensuring frequency splitting variations of less than 1 Hz. Theoretical modeling and simulation analysis indicated that the primary factors influencing the vibration modes and frequency splitting are the rectangular tooth structure and shell thickness. Following the optimized parameters, the frequency splitting of the All-Metal Micro Resonant Hemisphere was reduced by an order of magnitude to 14 Hz, demonstrating that these optimized conditions can significantly enhance the resonator’s performance.

## 1. Introduction

Due to its high precision, high stability, and low power consumption, the all-metal micro resonant hemisphere exhibits significant resistance to overload and magnetic interference. It can operate over a wide temperature range and adapt to various harsh environmental conditions, making it an indispensable core component in modern military equipment [1,2]. It plays a crucial role in the navigation, positioning, attitude control, inertial measurement, and other applications, enhancing the overall performance and reliability of military systems [3]. The hemispherical resonator, as a sensitive component of the all-metal micro resonant hemisphere, affects the accuracy and performance of the resonant gyroscope, and requires high machining precision [4,5].

During processing, inconsistencies may occur in the thickness of the hemispherical shell, as well as errors in the angle and position between the shell and the central axis rod, or surface defects, all of which can lead to the frequency splitting of the hemispherical resonator [6]. Traditional materials used for hemispherical resonators are mostly quartz, which is difficult to process and prone to defects. The sensitive component of the all-metal micro resonant hemisphere discussed in this paper is made from alloy materials, which offer better material uniformity, heat treatment adjustability, impact resistance, and vibration resistance, along with a superior corrosion resistance compared to quartz. Alloy materials typically a have better mechanical processing performance, making them easier to manufacture with high precision. This contributes to the improved manufacturing accuracy and consistency of the resonators.

In the working mode of n = 2, the two main axes of a hemispherical resonator possess their own resonant frequencies. Ideally, these two frequencies should be equal. However, due to shell thickness errors and various form and position errors during processing, the mass distribution and stiffness distribution on the hemispherical resonator change, resulting in a difference between the two frequencies—this is referred to as the frequency splitting of the hemispherical resonator [7,8].

Optimizing the design of the hemispherical resonator can streamline the process flow and reduce the unit production cost. The development trend favors small, lightweight, low-cost, high-precision, and mass-producible hemispherical resonators. Currently, Safran in France meets these criteria, with its products widely used in aerospace, defense technology, and in the industrial field [9]. The dynamic model of the hemispherical resonator serves as the foundation for gyroscope error analysis. Xu Zeyuan, Yi Guoxing, and others proposed a method for modeling the dynamics of a harmonic oscillator based on thin-shell theory in elastic mechanics. Through solving, they obtained the proportional coefficient and second-order resonant frequency expression of the harmonic oscillator, providing a computational basis for practical applications. This establishes a solid foundation for error analysis and the engineering applications of hemispherical resonators [10]. Yan Huo et al. established the motion equation of a hemispherical resonator with imperfect mass in the first harmonic of the second-order vibration mode. Through error mechanism analysis, they found that, when the linear vibration frequency equals the natural frequency of the resonator, the standing wave is constrained within the azimuth of different harmonics, with imperfect mass as the vibration excitation direction changes [11]. Wang Yuting and others improved the processing technology by analyzing the relationship between frequency splitting and quality error, reducing the frequency splitting to below 0.05 Hz [12]. Li Shaoliang and others obtained the influence law of standing wave swing effects on resonator frequency splitting through simulation experiments [13]. Tan Pinheng and others proposed ranges for wall thickness and a center shaft radius to ensure that the resonator exhibits better resistance to modal interference and impact overload [14]. Zheng Chao, Zhang Lin, and others optimized the shape and position errors of the hemispherical resonator and proposed design suggestions. Through finite element analysis, they identified the shape and position errors that significantly impact frequency splitting and quality factor, and conducted experimental verification [15]. Ruan Zhihu et al. provided a detailed analysis of the formation mechanism of tilt angle errors in the assembly of miniature hemispherical resonator gyroscopes, revealing the impact of the assembly tilt angle error on electrostatic driving capability, capacitance detection capability, and electrostatic stiffness adjustment capability [16].

This paper constructs the geometric model of a hemispherical resonator and analyzes the vibration modes of the all-metal micro resonant hemisphere. It examines the effects of different structural parameters on the natural frequency and frequency splitting, as well as the form and position errors that influence the frequency splitting of the resonator. Various error models are developed, and finite element simulation experiments are conducted to assess the impact of these errors on frequency splitting. The study provides recommended optimization ranges for the resonator, successfully reducing the frequency splitting error of the actual all-metal micro resonant hemisphere by an order of magnitude to 14 Hz. This research offers an experimental basis for the subsequent simulations and optimization experiments involving the resonator.

## 2. Working Principle and Geometric Modeling of the All-Metal Micro Resonant Hemisphere

### 2.1. Working Principle of the All-Metal Micro Resonant Hemisphere

The working mode of the harmonic oscillator is in the second order, specifically the four-wave antinode mode, which is an important vibration mode in all-metal micro resonant hemispheres [17]. As shown in Figure 1, the surface of the all-metal micro resonant hemisphere will exhibit four-wave antinodes and four-wave nodes, resulting in four alternating peak regions along the circumference. These peak regions are referred to as antinodes, where the surface displacement reaches its maximum, leading to the highest vibration amplitude of the all-metal micro resonant hemisphere. In contrast, a wave node is an area located between two adjacent antinodes, where the surface displacement is minimal or even zero.

In the four-wave antinode mode, there are also four-wave nodes uniformly distributed on the surface of the all-metal micro resonant hemisphere, located between the antinodes. The four antinodes typically exhibit a high degree of symmetry, with antinodes and nodes evenly distributed across the surface of the all-metal micro resonant hemisphere [18]. This symmetry helps reduce the influence of external interference on the vibration modes, thereby enhancing the stability and accuracy of the system. As shown in Figure 2, the two main frequencies of the four-wave antinode mode should be equal [19].

However, due to form and position errors, material inhomogeneity, or other defects in actual manufacturing, these two frequencies may experience frequency splitting, which is a significant factor affecting the accuracy of the all-metal micro resonant hemisphere [20]. In the four-wave antinode mode, there are typically two main vibration axes that pass through the center of the antinodes or near the nodes of the waves. These main axes are critical features that describe the vibration modes of the all-metal micro resonant hemisphere. The inherent rigid axis associated with the vibration mode usually defines the primary direction of vibration through the wave antinodes and wave nodes. These axes play a crucial role in analyzing form and position errors, as their precise positioning and orientation directly influence the modal frequency and overall system performance [21].

### 2.2. Geometric Structure of the All-Metal Micro Resonant Hemisphere

Based on previous studies on bell-shaped and hemispherical metal resonators, we present an all-metal micro resonator hemispherical structure featuring a rectangular tooth structure [22,23]. As shown in Figure 3 and Table 1, the structural parameters of a hemispherical resonator are as follows: hemispherical shell outer radius *r*, hemispherical shell thickness *h*, central axis rod diameter *d*, inner fillet radius r1, outer fillet radius r2, central axis rod length exceeding the equatorial plane of the hemispherical shell *L*, rectangular tooth thickness *H*, rectangular tooth length L1, and rectangular tooth spacing L2.

### 2.3. Natural Frequency and Frequency Splitting

The all-metal micro resonator hemispherical establishes its dynamic model based on the natural vibration theory of thin-walled spherical shells. Utilizing the Kirchhoff–Lyapunov hypothesis, the deformation analysis of thin-walled spherical shells is simplified. Any straight line perpendicular to the middle surface of the shell before deformation remains perpendicular to this surface after deformation, and the length of the normal line segment along the thickness of the shell remains constant during the deformation process. The normal stress generated between adjacent thin-shell layers parallel to the middle surface is very small compared to other components of the stress tensor and can be ignored [24]. In general, the deformation of the shell is represented by the combination of tangential displacement and normal displacement of the middle surface point [25]. The displacement of any point on the all-metal micro resonator hemispherical shell is:(1)V=u·a+v·b+w·c

Let *u*, *v*, and *w* be the displacement components of the point along the generatrix, loop, and normal directions of the all-metal micro hemispherical shell, respectively, while a, b, and c are the corresponding unit motion vectors. When the all-metal micro hemispherical shell is stationary, the n-th order symmetric mode of vibration in rotational space is expressed as:(2)$$\begin{array}{cc} u\left(\theta_{u},\theta_{v},t\right) & =u\left(\theta_{u}\right)\cos\theta_{v}\cos\omega t\\ v\left(\theta_{u},\theta_{v},t\right) & =v\left(\theta_{u}\right)\sin\theta_{v}\cos\omega t\\ w\left(\theta_{u},\theta_{v},t\right) & =w\left(\theta_{u}\right)\cos\theta_{v}\cos\omega t \end{array}$$

θu and θv are the coordinates of the all-metal micro hemispherical shell in the direction of the busbar and the direction of the loop, respectively.

Assuming that the all-metal micro hemispherical shell rotates at Ω=Ω1+Ω2 in inertial space, the expression in inertial space can be obtained:(3)$$\begin{array}{cc} \; & u\left(\theta_{u},\theta_{v},t\right)=u\left(\theta_{u}\right)\cos n\left(\theta_{v}+\theta \right)\cos\varphi_{n}t\\ \; & v\left(\theta_{u},\theta_{v},t\right)=v\left(\theta_{u}\right)\sin n\left(\theta_{v}+\theta \right)\cos\varphi_{n}t\\ \; & w\left(\theta_{u},\theta_{v},t\right)=w\left(\theta_{u}\right)\cos n\left(\theta_{v}+\theta \right)\cos\varphi_{n}t \end{array}$$
where θ is the precession angle, n is the circumferential wave number, φn is the resonant frequency, and $u\left(\theta u\right)$, $v\left(\theta u\right)$, and $w\left(\theta u\right)$ are the distribution patterns along the generatrix, loop, and normal directions, respectively.

Based on the above displacement formula, thin shell theory, and the vibration mode of the all-metal micro resonator hemispherical, we analyze the vibration bending state of the hemispherical surface:(4)$$\begin{array}{cc} \; & w\theta_{u}=-\frac{du\left(\theta_{u}\right)}{d\theta_{u}}\\ \; & nv\left(\theta_{u}\right)+u\left(\theta_{u}\right)\cos\theta_{u}=\frac{du\left(\theta_{u}\right)}{d\theta_{u}}\sin\theta_{u}\\ \; & nu\left(\theta_{u}\right)+v\left(\theta_{u}\right)\cos\theta_{u}=\frac{dv\left(\theta_{u}\right)}{d\theta_{u}}\sin\theta_{u} \end{array}$$

When the top boundary angle is small, the above equation can be approximated as:(5)$$\begin{array}{c} \frac{u\left(\theta_{u}\right)}{\sin\theta_{u}\tan^{n}\frac{\theta_{u}}{2}}=\frac{v\left(\theta_{u}\right)}{\sin\theta_{u}\tan^{n}\frac{\theta_{u}}{2}}=-\frac{w\left(\theta_{u}\right)}{\left(n+\cos\theta_{u}\right)\tan^{n}\frac{\theta_{u}}{2}} \end{array}$$

According to the vibration inertia force and work at any point on the all-metal micro resonator hemispherical, and based on the virtual displacement theorem, the natural frequency of the all-metal micro resonant hemisphere can be obtained:(6)$$\omega_{0}=\frac{k_{0}}{k_{1}}$$
(7)$$k_{0}=\frac{En^{2}\left(n^{2}+1\right)}{\rho \left(1+\mu \right){r_{1}}^{4}}\int_{\varphi_{0}}^{\varphi_{F}}\frac{\tan^{2n}\frac{\theta_{u}}{2}h}{3\sin^{3}\theta_{u}}d\theta_{u}$$
(8)$$k_{1}=\int_{\varphi_{0}}^{\varphi_{F}}\left(\sin^{2}\theta_{u}+2n\cos\theta_{u}+n^{2}+1\right)h\tan^{2n}\frac{\theta_{u}}{2}\sin\theta_{u}d\theta_{u}$$

One of the main sources of error in all-metal micro resonators hemispherical shells is the frequency splitting caused by the difference in natural frequencies between two vibration modes. The formula for the difference between the two frequencies is as follows:(9)$$\Delta \omega =\omega_{1}-\omega_{0}$$

$\omega_{1}$ is the higher order natural frequency of the same mode as $\omega_{0}$. The shape and position errors of the all-metal micro resonator hemispherical can lead to uneven mass distribution, resulting in the frequency splitting of its four-wave antinode working mode. This not only affects the angular velocity calculation accuracy of the all-metal micro hemispherical resonator gyroscope, but also has a negative impact on electrode assembly and excitation control.

## 3. Finite Element Analysis of All-Metal Micro Resonant Hemisphere

### 3.1. Grid Division Optimization

Draw a geometric model of the all-metal micro resonant hemisphere and perform grid division, as illustrated in Figure 4. The natural frequency under this uniform grid division is relatively low; however, the frequency difference in the four-wave antinode mode is significant. Therefore, the grid division is optimized for design purposes.

Due to the presence of rectangular tooth structures in all-metal Micro Resonant Hemispherical resonators, their geometric models are relatively complex, necessitating more precise mesh division to accurately capture standing wave modes. A detailed grid division can enhance the simulation accuracy of standing waves; however, overly detailed grids may significantly increase computational demands and affect overall efficiency.

Optimizing the grid distribution with a precision of 0.1 mm can maintain high accuracy in key areas where standing waves are captured, while allowing for reduced grid density in secondary areas, as shown in Figure 5. This approach achieves a balance between computational efficiency and accuracy. The stress concentration areas caused by standing waves typically require a finer mesh division to accurately simulate stress distribution. Therefore, grid optimization can appropriately increase the grid density based on the distribution characteristics of standing waves, particularly in high-stress regions.

The optimized grid design reduces the frequency splitting accuracy in the working mode from 10 Hz to 0.1 Hz while maintaining a stable natural frequency.

### 3.2. Simulation of Vibration Modes of the All-Metal Micro Resonant Hemisphere

The selected material is steel, known for its high yield strength and overload capacity, with a Young’s modulus *E* of 200,000 MPa, a material density ρ of 7.85 g/mm³, a Poisson’s ratio μ of 0.3, a thermal expansion coefficient of 0.000012/℃, a compressive yield strength of 250 MPa, and a tensile yield strength of 250 MPa. Finite element simulations were performed, and the first to tenth order vibration modes are shown in Figure 6.

The first and second orders of the all-metal micro hemispherical resonator correspond to the working modes, with a 45° difference in vibration mode, as shown in Table 2. The natural frequencies are 10,816.27224 Hz and 10,816.79813 Hz, resulting in a frequency difference of 0.52589 Hz. The third and fourth orders represent the left and right oscillation modes of the hemispherical shell; the fifth and sixth orders correspond to the six-wave antinode modes; the seventh order denotes the up and down vibration mode of the resonator; the eighth and ninth orders are associated with the eight-wave antinode modes; and the tenth order indicates the left and right rotation mode of the all-metal micro hemispherical resonator.

### 3.3. Structural Parameter Simulation of the All-Metal Micro Resonant Hemisphere

Conduct structural parameter simulation experiments on the all-metal micro hemispherical resonators, with the simulation range and step size presented in Table 3. The double bars in the table indicate that the geometric parameters are fixed parameters.

The size of the central axis rod that exceeds the length of the equatorial plane of the all-metal micro hemispherical shell has little effect on the natural frequency and frequency splitting, and can therefore be ignored. Thus, through finite element simulation, structural parameters such as outer radius, shell thickness, central axis rod diameter, inner fillet radius, outer fillet radius, as well as the length, thickness, and spacing of the rectangular teeth were varied while maintaining the same mesh parameters. Equal step simulations were conducted, and the simulation results are presented below.

As shown in Figure 7, the natural frequency gradually decreases as the radius of the all-metal micro hemispherical shell increases, with the highest sensitivity observed in the shell oscillation mode. The frequency splitting remains relatively small around a radius of 2.9 mm.

As shown in Figure 8, the natural frequency gradually increases with an increase in shell thickness. The frequency splitting is relatively small around a thickness of 0.1 mm.

The changes in *r* and *h* have a significant impact on the natural frequency, while variations in *d*, r1, and r2 have the greatest effect on the shell oscillation mode, as shown in Figure 9, Figure 10 and Figure 11. *r* has the most substantial impact on the variation of the modal order and frequency splitting, whereas *h* significantly affects the frequency difference of higher-order modes. The frequency splitting of the working mode near the fundamental value is relatively small, generally less than 1 Hz.

As shown in Figure 12, Figure 13 and Figure 14, the length of the rectangular teeth has the greatest impact on the natural frequency, while the spacing of the teeth has the least effect. As the thickness of the teeth increases, particularly when it exceeds the baseline value, the change in low-order natural frequency becomes smaller. The frequency difference in low-order modes is relatively minor, typically less than 1 Hz. Changes in structural parameters that significantly affect quality can lead to substantial frequency splitting.

## 4. Factors Affecting Frequency Splitting of the All-Metal Micro Resonant Hemisphere

### 4.1. Rectangular Tooth Angle Error

The rectangular teeth of the all-metal micro hemispherical resonator are designed to increase the contact area between the resonator and the base plane electrode, ensuring effective excitation and detection by the electrode. During processing, the shape and size errors of the rectangular teeth can impact the resonator’s contact performance and the accuracy of the electrode signal. As shown in Figure 15, the rectangular teeth may exhibit an angular error relative to the hemispherical shell during processing, leading to uneven contact between the electrodes and affecting the excitation efficiency and signal detection stability of the resonator. The rectangular tooth error in this case is referred to as the forward error.

As shown in Table 4, the error parameter range is set from −0.1° to 0.1°, with a simulation step size of 0.01°.

We performed finite element simulations for the first to tenth orders and analyzed the four modes that generate frequency differences: namely, the four-wave antinode mode, hemispherical shell sway mode, six-wave antinode mode, and eight-wave antinode mode, as shown in Figure 16. The experimental results indicate that the frequency differences in these four modes are relatively small near 0°, generally less than 1 Hz. When the error is positive, the change in frequency splitting is minimal; however, when the error is negative, the change is substantial. A negative error significantly impacts the natural frequency. Due to the negative error, the distance between the rectangular teeth and the electrode plate is reduced, which may lead to collisions between the rectangular teeth and the electrode plate as a result of changes in the vibration mode.

### 4.2. Position Error of the Central Axis Rod and the All-Metal Micro Resonant Hemisphere

In the manufacturing process of the all-metal micro hemispherical resonator, angle and displacement errors may occur between the central axis of the resonator housing and the central axis of the central shaft [26]. As shown in Figure 17, the angle error refers to the angular deviation between the central axis of the resonator shell and the central axis of the central shaft. This error can shift the vibration mode of the resonator shell, disrupting the symmetry of the four-wave antinode modes and leading to frequency splitting. The displacement error is the horizontal deviation between the axis of the resonator shell and the axis of the central shaft, which can result in uneven mass distribution in the resonator shell and affect its vibration mode.

An angle error may occur between the central axis of the central axis rod and the central axis of the hemispherical shell, referred to as the central axis angle error. As shown in Table 5, the simulation is conducted within the range of 0° to 0.1°, with a simulation step size of 0.01°, using the ideal situation as a reference. An increase in the angle error will correspondingly increase frequency splitting. As shown in Figure 18 and Figure 19, the natural frequency and frequency splitting under the four-wave antinode mode will rise with an increase in error, with the maximum frequency difference not exceeding 3.5 Hz. During the manufacturing process, it is crucial to strictly control the angle error between the resonator shell and the central shaft, typically maintaining it within 0.1°.

There will be a horizontal position error between the two central axes, which will cause the vibration mode to shift at the center, as illustrated in the four-wave antinode mode and shell oscillation mode in Figure 20. The red line shows the original four-antinode shape. The center of the vibration mode will change in the direction of the center of gravity shift, and the natural frequency will decrease as the error increases. The frequency splitting will also decrease with an increase in error, with the error having the greatest impact on frequency splitting in the shell oscillation mode.

### 4.3. Thickness Error of Hemispherical Shell

The shell thickness error caused by the offset of the center position of the all-metal micro hemispherical resonator will affect its structural symmetry [27]. This shell thickness error can lead to changes in the local mass distribution and stiffness of the all-metal micro hemispherical resonator, which directly impacts the resonant frequency of the resonator. The resonator relies on a symmetrical structure to achieve precise resonance modes, while uneven shell thickness results in symmetry failure, leading to inconsistent vibration mode frequencies at different positions [28].

The shell thickness error is primarily caused by the offset of the inner circle center relative to the outer circle center. This center offset occurs vertically along the central axis, as illustrated in Figure 21. The left figure depicts the center moving downward along the central axis, resulting in an increase in local shell thickness, while the right figure shows an upward movement, leading to a decrease in local shell thickness.

The shell thickness error is attributed to the offset of the center position. As shown in Table 6, the finite element simulation range is set from −0.05 mm to 0.05 mm, with a simulation step size of 0.005 mm. As shown in Figure 22, starting from a downward displacement of 0.05 mm from the center of the circle, the resonant mass increases due to the gradual thickening of the shell. Consequently, the natural frequencies of the three modes rise as the center of the circle moves upward. However, due to the reduction in shell thickness and uneven local mass distribution, the frequency splitting of the four-wave antinode mode and shell oscillation mode, caused by the downward movement of the center of the circle is greater.

## 5. Optimization Design of the All-Metal Micro Hemispherical Resonator

Based on the changes in natural frequency and frequency splitting observed during the structural parameter simulations, corresponding optimization suggestions are provided for each parameter to ensure that the frequency difference does not exceed 1 Hz. Please refer to Table 7 for optimization parameters.

According to the frequency differences caused by size changes, it can be observed that the machining of rectangular teeth during actual processing is prone to errors, resulting in structural asymmetry and uneven quality. Adjustments to the three structural parameters L1, L2, and *r* can lead to significant frequency differences. The size adjustments of *r*, *h*, and *d* are more likely to cause changes in modal order. Notably, the size adjustments of *h* and $L_{1}$ result in significant changes in the natural frequency. The processing of shell thickness *h* and rectangular teeth is relatively fine, making it susceptible to errors and uneven local quality.

## 6. Test Experiment of All-Metal Micro Hemispherical Resonator

After optimizing the all-metal micro resonator hemispherical according to the structural parameters mentioned above, a laser sweep test experiment is conducted. The actual four-antinode vibration frequency of the optimized resonator, obtained from the test, is 11,547 Hz. Compared to the simulation results, the error is 0.0675, which demonstrates that the results obtained through finite element analysis in this paper are reliable.

As shown in Figure 23, this schematic diagram illustrates the test points for the all-metal micro resonant hemisphere. The frequency sweep results indicate that the resonant frequency of the actual resonator is significantly reduced after optimization, with the frequency splitting decreased to 14 Hz. Considering factors such as the processing and material non-uniformity of the resonator, this value represents the lower frequency splitting of the existing all-metal micro resonant hemisphere.

## 7. Conclusions

This paper provides optimization suggestions for the structural parameters of the all-metal micro resonator hemispherical shell based on finite element simulations. The structural parameters for the hemispherical shell are as follows: *r* ranging from 2.8 to 3.0 mm, *h* from 0.09 to 0.10 mm, *d* from 1.3 to 1.5 mm, $r_{1}$ from 0.8 to 0.9 mm, $r_{2}$ of 1.0 mm, *L* of 0.1 mm, *H* from 0.09 to 0.10 mm, $L_{1}$ from 0.9 to 1.0 mm, and $L_{2}$ from 0.14 to 0.15 mm.

A simulation analysis was conducted on potential rectangular tooth angle errors, central axis rod, and hemispherical shell position errors, and shell thickness errors that may occur during the manufacturing process. The experimental structure indicated that, when the rectangular tooth angle error is negative, it significantly impacts the natural frequency and frequency splitting. When the error approaches 0°, the frequency splitting remains relatively small, typically less than 1 Hz. Angle errors and displacement errors between the central axes can cause displacements and symmetry disruptions in the vibration mode, leading to increased frequency splitting. The maximum frequency difference under the four-wave antinode mode, within the ranges of a 0.1° angle error and a 0.055 mm displacement error, can reach 2 Hz. This indicates that, in actual manufacturing processes, it is essential to strictly control the angle error between the resonator shell and the central shaft to avoid significant frequency splitting. The horizontal position error can lead to a shift in the center of vibration modes, particularly in the four-wave antinode mode and shell oscillation mode. Due to the offset of the center of gravity, the center of the vibration mode will change accordingly, with the shell oscillation mode being most sensitive to frequency decomposition associated with this horizontal position error. The simulation also analyzed the shell thickness error caused by the offset of the center position. The results showed that increasing the shell thickness would improve the quality of the resonator, leading to an increase in the natural frequency. Conversely, a decrease in shell thickness results in uneven local mass; when the center of the circle moves downward, the frequency splitting of the four-wave antinode mode and shell oscillation mode will significantly increase.

The optimized resonator’s resonant frequency exhibits a deviation of 6.75 compared to the simulation results, demonstrating the reliability of the finite element analysis conducted in this study. The optimized frequency splitting is 14 Hz, which is a relatively low value for metal materials. In the design and manufacturing process of the resonator, the design of structural parameters is crucial, and it is essential to strictly control the angle error, horizontal position error, and shell thickness error between the shell and the central axis rod. Specifically, the angle error should be maintained within 0.1° to minimize the increase in frequency splitting. Controlling the position of the center is vital to avoid modal frequency differences and vibration offsets caused by uneven mass distribution. This research provides an important reference value for improving the design and manufacturing precision of resonant gyroscopes.

## Figures and Tables

**Figure 1 sensors-24-07132-f001:**
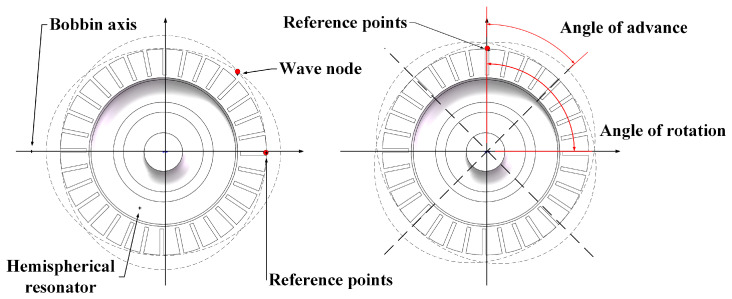
Diagram of four-wave antinodes for the all-metal micro resonant hemisphere.

**Figure 2 sensors-24-07132-f002:**
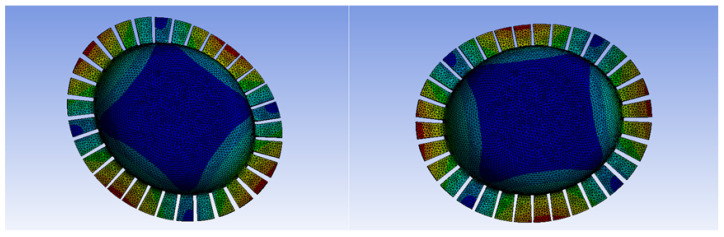
Simulation of four antinode modes of the all-metal micro resonant hemisphere.

**Figure 3 sensors-24-07132-f003:**
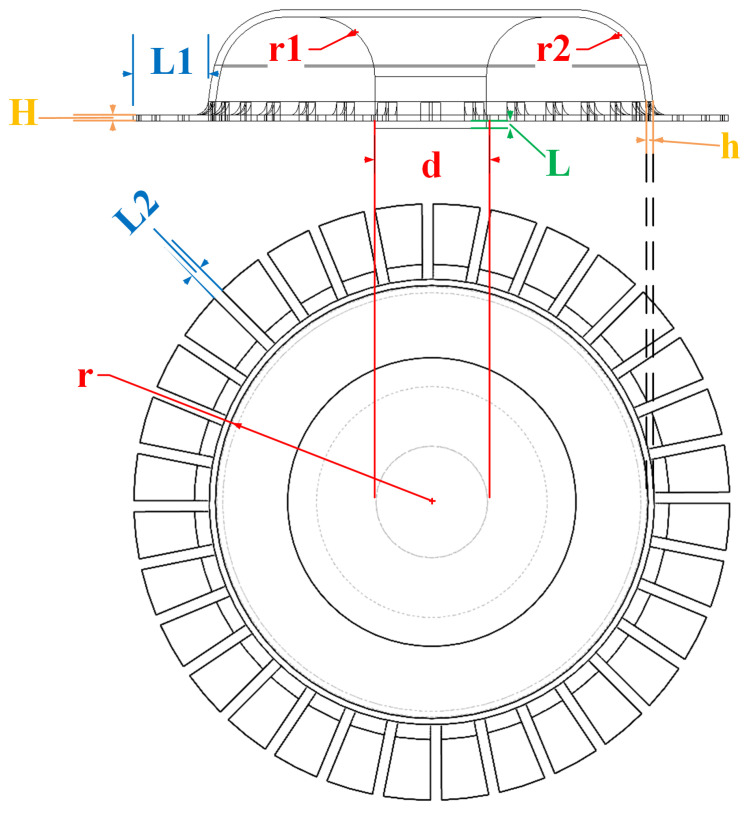
Parameter annotation diagram of the all-metal micro resonant hemisphere structure.

**Figure 4 sensors-24-07132-f004:**
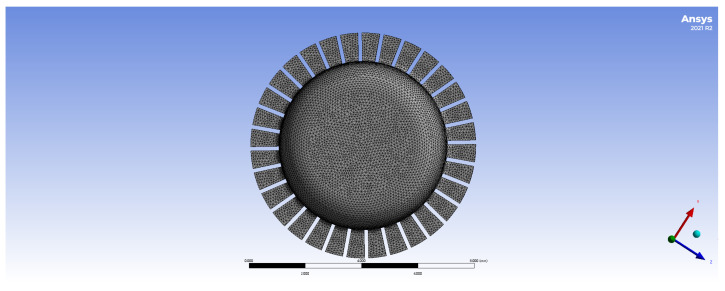
Grid division diagram of the all-metal micro resonant hemisphere.

**Figure 5 sensors-24-07132-f005:**
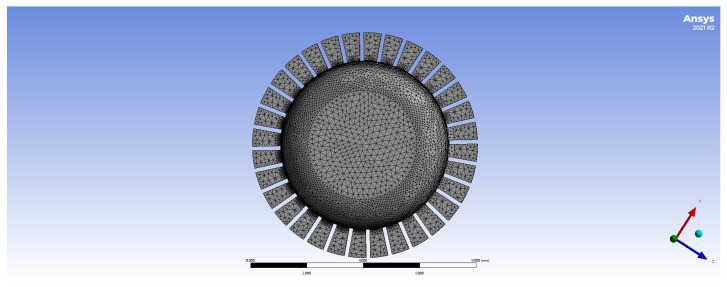
Optimization diagram of the all-metal micro resonant hemisphere grid.

**Figure 6 sensors-24-07132-f006:**
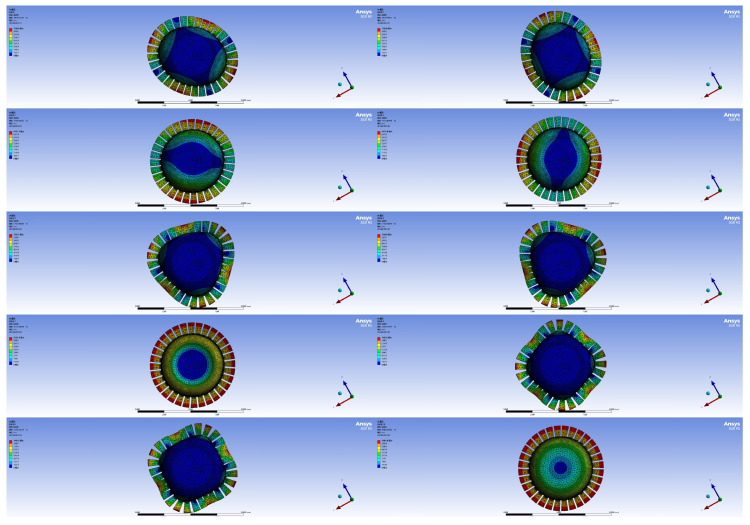
Tenth-order resonator simulation diagram.

**Figure 7 sensors-24-07132-f007:**
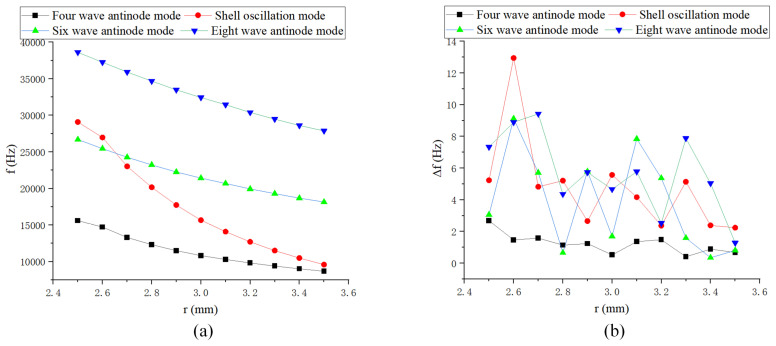
Half spherical shell radius simulation: (**a**) Natural frequency curve; and (**b**) Frequency decomposition curve.

**Figure 8 sensors-24-07132-f008:**
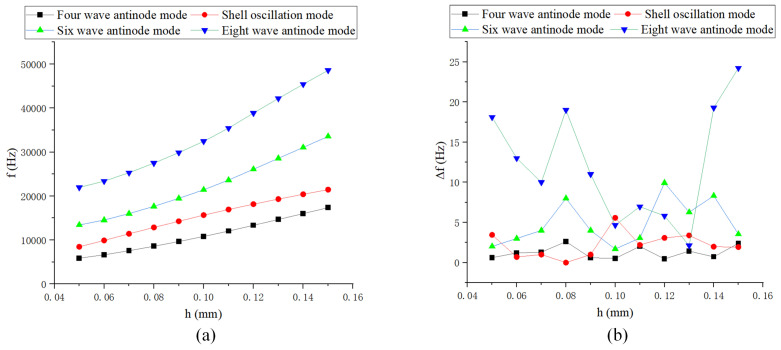
Simulation of hemispherical shell thickness: (**a**) Natural frequency curve; and (**b**) Frequency decomposition curve.

**Figure 9 sensors-24-07132-f009:**
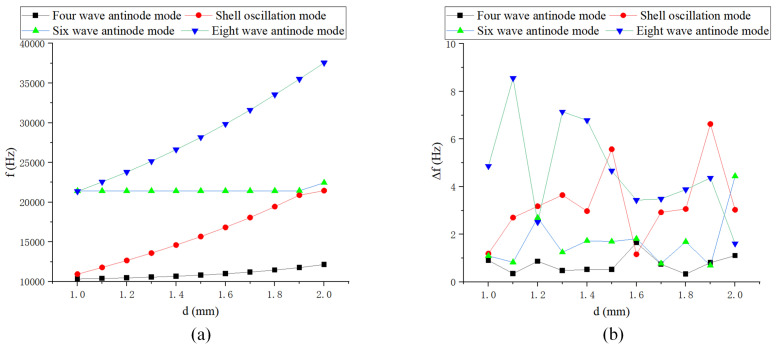
Simulation of the diameter of the central axis rod: (**a**) Natural frequency curve; and (**b**) Frequency decomposition curve.

**Figure 10 sensors-24-07132-f010:**
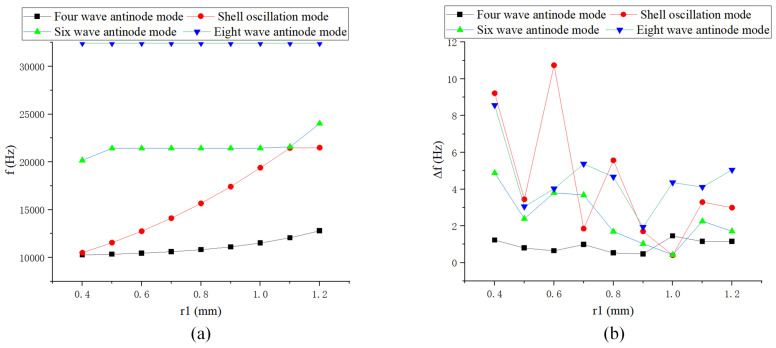
Simulation of inner fillet radius: (**a**) Natural frequency curve; and (**b**) Frequency decomposition curve.

**Figure 11 sensors-24-07132-f011:**
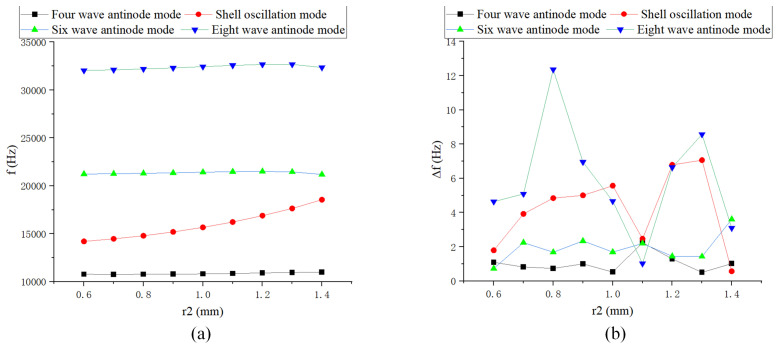
Simulation of outer corner radius: (**a**) Natural frequency curve; and (**b**) Frequency decomposition curve.

**Figure 12 sensors-24-07132-f012:**
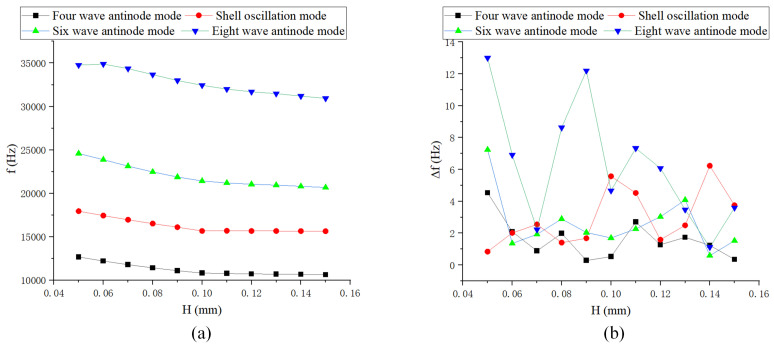
Simulation of rectangular teeth thickness: (**a**) Natural frequency curve; and (**b**) Frequency decomposition curve.

**Figure 13 sensors-24-07132-f013:**
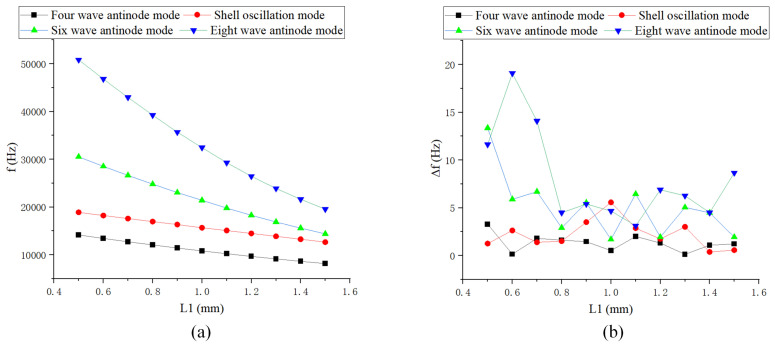
Simulation of rectangular teeth length: (**a**) Natural frequency curve; and (**b**) Frequency decomposition curve.

**Figure 14 sensors-24-07132-f014:**
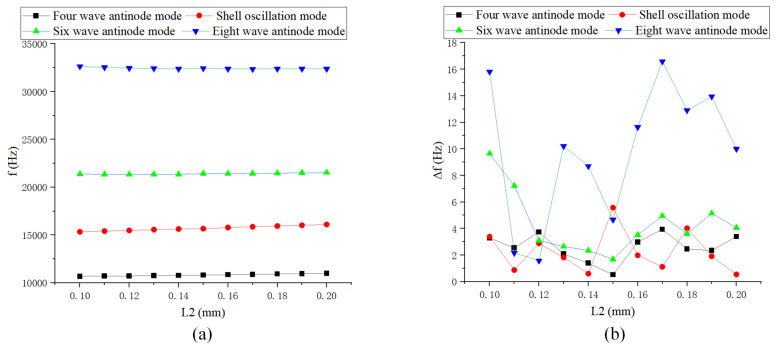
Simulation of rectangular teeth spacing: (**a**) Natural frequency curve; and (**b**) Frequency decomposition curve.

**Figure 15 sensors-24-07132-f015:**
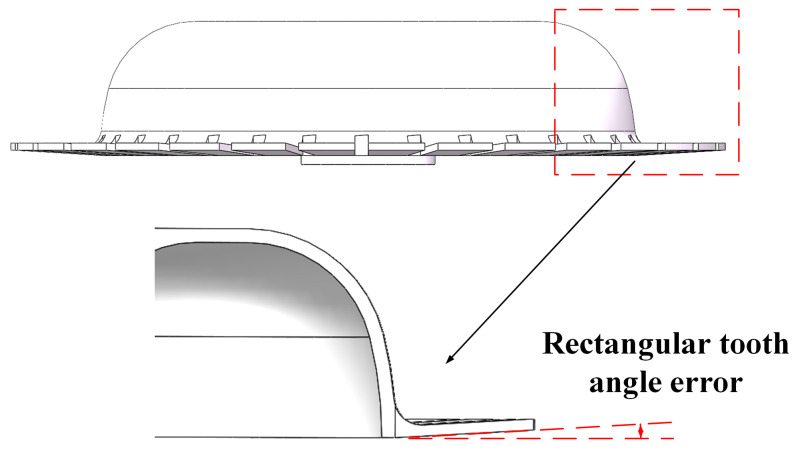
Error diagram of rectangular tooth angle.

**Figure 16 sensors-24-07132-f016:**
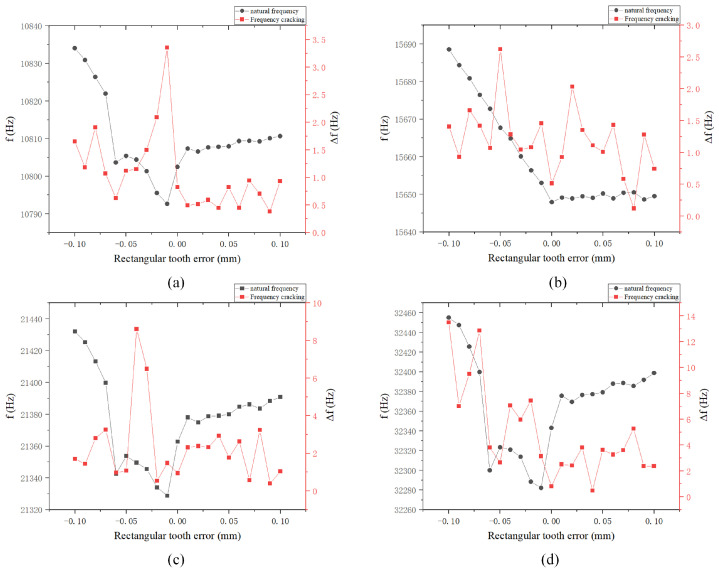
Natural frequency and frequency decomposition simulation of rectangular tooth error: (**a**) Four-wave antinode mode; (**b**) Shell oscillation mode; and (**c**) Six-wave antinode mode; and (**d**) Eight-wave antinode mode.

**Figure 17 sensors-24-07132-f017:**
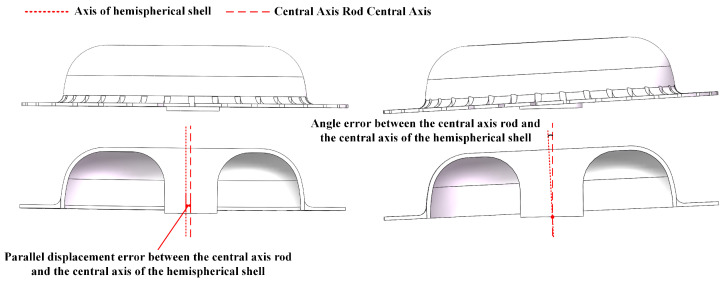
Error diagram of the central axis rod and the all-metal micro hemispherical resonator.

**Figure 18 sensors-24-07132-f018:**
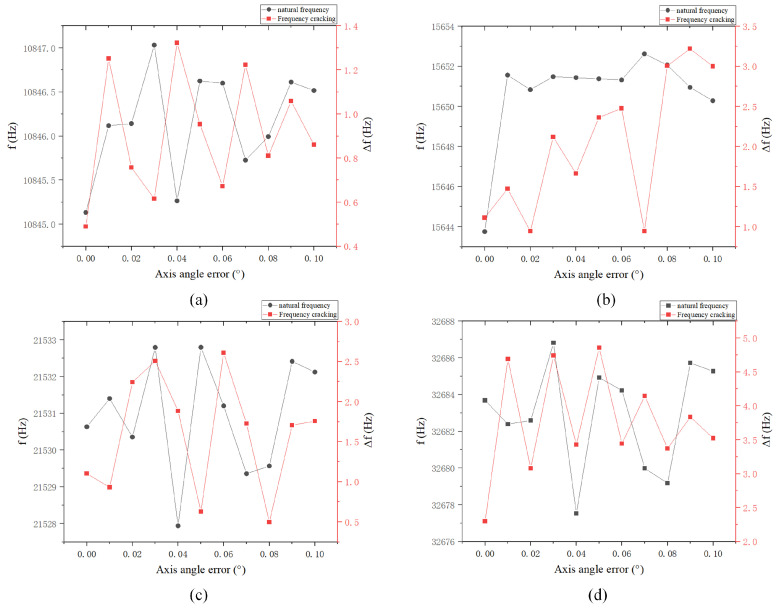
Natural frequency and frequency decomposition simulation of the angle error between the central axis rod and the hemispherical shell: (**a**) Four-wave antinode mode; (**b**) Shell oscillation mode; (**c**) Six-wave antinode mode; and (**d**) Eight-wave antinode mode.

**Figure 19 sensors-24-07132-f019:**
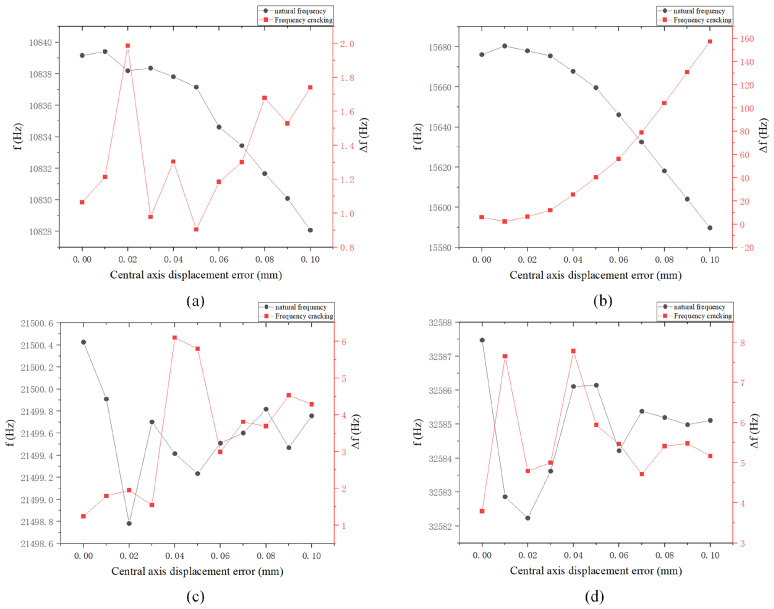
Natural frequency and frequency decomposition simulation of displacement error between central axis rod and hemispherical shell: (**a**) Four-wave antinode mode; (**b**) Shell oscillation mode. (**c**) Six wave antinode mode; and (**d**) Eight wave antinode mode.

**Figure 20 sensors-24-07132-f020:**
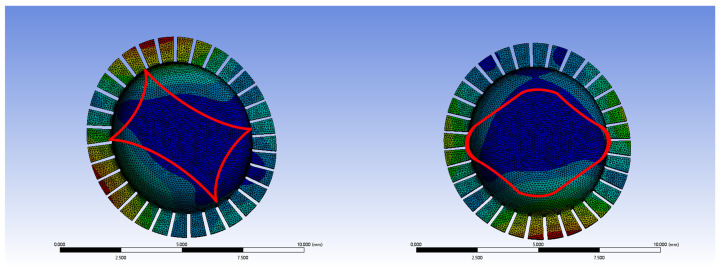
Vibration mode shift of the four-wave antinode mode and shell oscillation mode.

**Figure 21 sensors-24-07132-f021:**
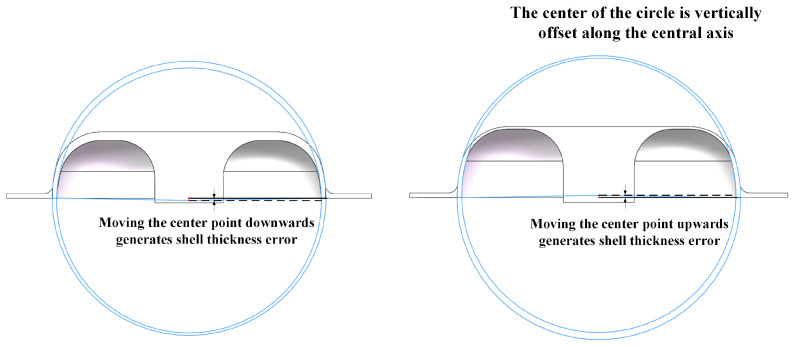
Diagram of shell thickness error caused by center point offset.

**Figure 22 sensors-24-07132-f022:**
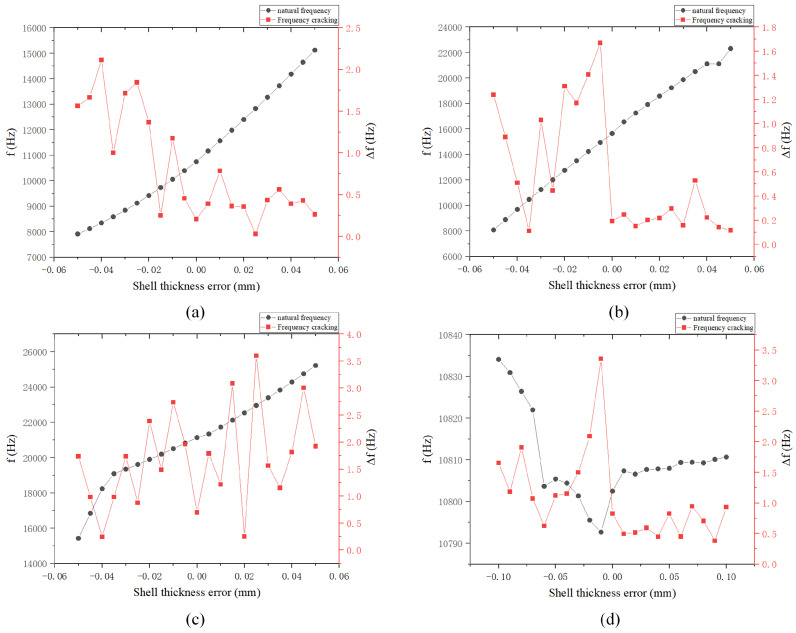
Natural frequency and frequency splitting variation diagram under shell thickness error: (**a**) Four-wave antinode mode; (**b**) Shell oscillation mode; (**c**) Six-wave antinode mode; and (**d**) Eight-wave antinode mode.

**Figure 23 sensors-24-07132-f023:**
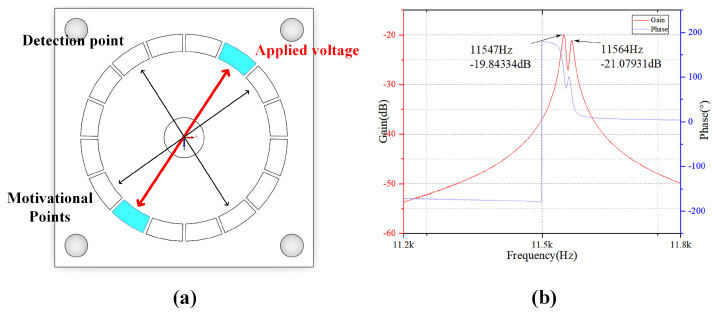
Test experiment diagram: (**a**) Test point annotation diagram; and (**b**) Sweep frequency graph.

**Table 1 sensors-24-07132-t001:** Table of structural and physical parameters of the all-metal micro resonant hemisphere.

Parameters	Symbol	Basic Value (mm)
Outer radius of the hemispherical shell	*r*	3.0
Thickness of the hemispherical shell	*h*	0.1
Diameter of the central axis rod	*d*	1.5
Inner corner radius	$r_{1}$	0.8
Outer corner radius	$r_{2}$	1.0
Length of the central axis rod exceeding the equatorial plane of the hemispherical shell	*L*	0.1
Thickness of the rectangular teeth	*H*	0.1
Length of the rectangular teeth	$L_{1}$	1.0
Spacing between rectangular teeth	$L_{2}$	0.15

**Table 2 sensors-24-07132-t002:** Natural frequency table of the first-to-tenth order resonators.

Order	Frequency (Hz)	Modal
1	10,816.27224	Four-wave antinode mode
2	10,816.79813	Four-wave antinode mode
3	15,666.46362	Shell sway mode
4	15,672.03469	Shell sway mode
5	21,424.60906	Six-wave antinode mode
6	21,426.30646	Six-wave antinode mode
7	28,173.50598	Upper and lower vibration modes of hemispherical shell
8	32,441.61547	Eight-wave antinode mode
9	32,446.28254	Eight-wave antinode mode
10	40,404.58662	Hemisphere shell left and right rotation mode

**Table 3 sensors-24-07132-t003:** Simulation parameter table.

Structural Parameters	Basic Value (mm)	Simulation Scope (mm)	Simulation Step Size (mm)
*r*	3.0	2.5–3.5	0.1
*h*	0.1	0.05–0.15	0.01
*d*	1.5	1–2	0.1
$r_{1}$	0.8	0.4–1.2	0.1
$r_{2}$	1.0	0.6–1.4	0.1
*L*	0.1	//	//
*H*	0.1	0.05–0.12	0.01
$L_{1}$	1.0	0.5–1.5	0.1
$L_{2}$	0.15	0.1–0.2	0.01

**Table 4 sensors-24-07132-t004:** Simulation parameter table for rectangular tooth angle error.

Error	Basic Value	Simulation Scope	Simulation Step Size
Rectangular tooth angle error	0°	−0.1–0.1°	0.01°

**Table 5 sensors-24-07132-t005:** Simulation parameter table for the position error of the central axis rod and hemispherical shell.

Error	Basic Value	Simulation Scope	Simulation Step Size
Angle error between the central axis	0°	0.01–0.1°	0.01°
Parallel displacement error between the central axis	0 mm	0.01–0.055 mm	0.005 mm

**Table 6 sensors-24-07132-t006:** Simulation parameter table for shell thickness error.

Error	Basic Value	Simulation Scope	Simulation Step Size
Shell thickness error	0 mm	−0.05–0.05 mm	0.005 mm

**Table 7 sensors-24-07132-t007:** Structural parameter optimization table.

Structure Parameters	Optimize Parameters (mm)	Δf (Hz)
*r*	2.8–3.0	0.614
*h*	0.09–0.10	0.074
*d*	1.3–1.5	0.048
$r_{1}$	0.8–0.9	0.051
$r_{2}$	1.0	//
*L*	0.1	//
*H*	0.09–0.10	0.242
$L_{1}$	0.9–1.0	0.929
$L_{2}$	0.14–0.15	0.876

## Data Availability

The data presented in this study are available in the article.

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
