# Peer review of "Research on the Configuration Optimization of All-Metal Micro Resonant Hemisphere"

_sensors, 2024, doi:10.3390/s24227132_

Round 1
Reviewer 1 Report
Comments and Suggestions for Authors
This manuscript has many technical mistakes and it lacks scientific contribution or novelty for publication in a peer-reviewed journal. This manuscript does not have an acceptable organization and scientific writing. The title is incorrect. This manuscript does not incorporate a formal optimization process. This manuscript only adds the results of simulations of a resonator, which are not validated with experimental results. The abstract is not poor. The introduction did not include a research problem. In addition, this section did not present a scientific contribution or novelty. This manuscript did not report a detailed description of the design of the proposed resonator. Furthermore, the design of the resonator must be based on the design rules of a fabrication process. Also, this manuscript did not present the fabrication and experimental results of the resonator. The experimental results are needed to validate the simulation results. The discussions of the main results are poor and weak. Figures 3-22 are not acceptable for publication in a scientific paper.
Comments on the Quality of English LanguageEnglish grammar and style must be improved.
Author Response
Comment 1: The title is incorrect.
Response 1: We accept your comments and have changed the title of the article to Research on Configuration Optimization of All-metal Micro Resonant Hemisphere.
Comment 2: This manuscript does not incorporate a formal optimization process. This manuscript only adds the results of simulations of a resonator, which are not validated with experimental results.
Response 2: We appreciate your comments and have revised the title of the article accordingly. Additionally, we have included the experimental section in Chapter 6 to validate the optimization. The resonant frequency error is minimal, which confirms the validity of the simulation results. You can find the modified content in line 304.
Comment 3: The introduction did not include a research problem.
Response 3: Thank you for your valuable feedback regarding the introduction. We acknowledge your suggestion and have made the necessary modifications by including a description of the research problem in the introduction. You can find the updated content in line 76 of the manuscript.
Comment 4: This manuscript did not report a detailed description of the design of the proposed resonator.
Response 4: Thank you for your constructive feedback regarding the description of the resonator design. We acknowledge your suggestion and have made the necessary revisions. This study is primarily based on previous research, and we have added relevant citations to demonstrate that the design of the resonator is grounded in the design rules of the fabrication process. You can find the updated content in line 114 of the manuscript.
Comment 5: The experimental results are needed to validate the simulation results. The discussions of the main results are poor and weak.
Response 5: Thank you for your valuable feedback regarding the need for experimental results to validate the simulation findings. We acknowledge your suggestion and have made the necessary revisions. We have conducted actual experiments to verify the optimization results from the simulations, and you can find this information in line 310 of the manuscript. Additionally, we have enhanced the discussion of the main results and included a detailed description of these results in the conclusion. You can find the updated content in line 344.
Comment 6: Figures 3-22 are not acceptable for publication in a scientific paper.
Response 6: Thank you for your constructive feedback regarding the figures in the manuscript. We acknowledge your concerns and have made the necessary revisions. The figures have been reformatted and re-uploaded to ensure they meet the publication standards for a scientific paper. You can view the updated figures above line 284 in the revised manuscript.

Reviewer 2 Report
Comments and Suggestions for Authors
Comments on: Research on Configuration Optimization of All-metal MicroHemisphere Resonant
The authors carry out a systematic numerical analysis to determine the sensitivity of the resonant frequencies of the vibration modes of a hemispherical-shaped resonator to various parameters. The simulation results are presented in Figs 7-14. The authors conclude that the rectangular tooth length is the most important parameter and that errors in the angle that the tooth makes can cause the resonant frequencies of the two desired modes to split from each other.
The English is poor throughout, including the title itself which should read: "Research on Configuration Optimization of All-metal Micro ResonantHemisphere"
Another example in line 221: "Perform one to ten order finite element simulations on it, and analyze the four modes that generate frequency differences, namely four wave antinode mode, hemispherical shell sway mode, six antinode mode, and eight antinode mode."
This sentence apparently should be: "We performed one to ten order finite element simulations ..."
There are many similar examples of poor English throughout.
One question: in line 171, the thermal expansion coefficient is listed as being 120,000 per degree C? This seems to be high by many orders of magnitude.
Summary: This appears to be a well thought-out study that is worthy of publication once the English is improved. The abstract, introduction, and references appear appropriate.
Comments on the Quality of English LanguageThe quality of the English is poor throughout, including the title (which should end with "Micro Resonant Hemisphere")
Author Response
Comment 1: The English is poor throughout, including the title itself which should read: "Research on Configuration Optimization of All-metal Micro Resonant Hemisphere."
Response 1: Thank you for your feedback regarding the language quality throughout the manuscript. I acknowledge this point and have made the necessary revisions. Firstly, I have thoroughly reworked the English translation of the entire text to enhance its clarity and coherence. Additionally, I have modified the title of the paper according to your suggestion, which you can find in the first line of the manuscript.
Comment 2: "Perform one to ten order finite element simulations on it, and analyze the four modes that generate frequency differences, namely four wave antinode mode, hemispherical shell sway mode, six antinode mode, and eight antinode mode." This sentence apparently should be: "We performed one to ten order finite element simulations..." There are many similar examples of poor English throughout.
Response 2: Thank you for your valuable feedback regarding the language quality in the manuscript. I acknowledge this point and have made the necessary revisions. I have corrected the issue with the sentence you mentioned, changing it to "We performed one to ten order finite element simulations..." as per your suggestion. You can find these modifications reflected in the text. Specifically, the revised sentence appears in line 235 of the manuscript.
Comment 3: In line 171, the thermal expansion coefficient is listed as being 120,000 per ℃. This seems to be high by many orders of magnitude.
Response 3: Thank you for your insightful feedback regarding the thermal expansion coefficient listed in line 171. I acknowledge this point and have made the necessary revision. The previously stated value of 120,000 per ℃ was indeed incorrect, and I have corrected it to a more accurate figure. You can find the revised value in line 184 of the manuscript.

Round 2
Reviewer 1 Report
Comments and Suggestions for Authors
This version of the manuscript was improved based on the reviewer's comments.
Comments on the Quality of English LanguageThe English grammar can be improved.
Reviewer 2 Report
Comments and Suggestions for Authors
The authors have corrected some things and the overall improvement in the English is substantial. Thus I believe that the manuscript is acceptable for publication in its present form.